

# Rainfall erosivity factor in the Czech Republic and its Uncertainty

Martin Hanel[1,2], Petr Máca[1], Petr Bašta[1], Radek Vlnas[1], and Pavel Pech[1]

[1]Faculty of Environmental Sciences, Czech University of Life Sciences, Kamýcká 1176, Prague 6, Czech Republic
[2]T. G. Masaryk Water Research Institute, Podbabská 30, Prague 6, Czech Republic

*Correspondence to:* Martin Hanel (hanel@fzp.czu.cz)

**Abstract.** In the present paper, the rainfall erosivity factor (R-factor) for the area of the Czech Republic is assessed. Based on 10-minute data for 96 stations and corresponding R-factor estimates, a number of spatial interpolation methods are applied and cross-validated. These methods include inverse distance weighting, standard, ordinary and regression kriging with parameters estimated by the method of moments and restricted maximum likelihood and a generalized least-squares (GLS) model. For the regression-based methods, various statistics of monthly precipitation as well as geographical indices are considered as covariates. In addition to the uncertainty originating from spatial interpolation, also the uncertainty due to estimation of the rainfall kinetic energy (needed for calculation of the R-factor) as well as the effect of record length and spatial coverage are addressed. Finally, the contribution of each source of uncertainty is quantified. The average R-factor for the area of the Czech Republic is 64 MJ ha$^{-1}$ cm h$^{-1}$, with values for the individual stations ranging between 32 and 152 MJ ha$^{-1}$ cm h$^{-1}$. Among various spatial interpolation methods, the GLS model relating R-factor to the mean altitude, longitude, mean precipitation and mean excess above the 95th percentile of monthly precipitation performed best. Application of the GLS model also reduced the uncertainty due to the record length, which is substantial when the R-factor is estimated for individual sites. Our results revealed that reasonable estimates of the R-factor can be obtained even from relativelly short records (15–20 years), provided sufficient spatial coverage and covariates are available.

## 1 Introduction

Erosion is a natural geological phenomenon resulting from the removal of soil particles by water or wind. Soil erosion by water is a widespread problem throughout Europe (Van der Knijff et al., 2000). Erosion is usually triggered by a combination of factors like climate (e.g. long dry periods followed by heavy rainfall), topography (steep slopes), inappropriate land use, land cover patterns (e.g. sparse vegetation), ecological disasters (e.g. forest fires) and soil characteristics (e.g. a thin layer of topsoil, silty texture or low organic matter content).

Although measurements of soil erosion exist, they are often used as a basis for development, modification or verification of soil erosion models (applicable at larger scales), which are relating soil loss to indicators of relevant factors. A classic example is the Universal soil loss equation (USLE, Wischmeier and Smith, 1978) or the Revised USLE (Renard et al., 1997). Both methods express the long-term average annual soil loss as a product of rainfall erosivity factor (R-factor), soil erodibility factor, slope length factor, slope steepness factor, cover-management factor and support practice factor. The experimental data indicate that when factors other than rainfall are held constant, soil losses from cultivated fields are directly proportional to an



erosivity index ($EI30$) calculated as the total rainfall kinetic energy times the maximum 30-min intensity (Renard et al., 1997). The R-factor is then obtained as a long-term average annual rainfall erosivity index.

Because of large spatiotemporal variability of rainfall, long records from a dense network of stations are in general required in order to provide reliable estimates of the R-factor and/or to develop rainfall erosivity maps. On the other hand, lack of

5 high-resolution rainfall data in combination with a need for soil erosion risk assessment often leads to situations where the R-factor is estimated based on relatively short records. For instance, many of the stations used recently for the derivation of rainfall erosivity maps for Europe (Panagos et al., 2015) were shorter than 20 years and even a number of records shorter than 10 years were considered. Similarly, the comparison of the spatial interpolation methods in the Ebro Basin (Angulo-Martínez et al., 2009) was based on 10 years of data (for a large number of stations) and Catari et al. (2011) assessed the uncertainty in

the estimated R-factor considering a 13-year record for eight stations. These record lengths are significantly shorter than the 22–30 years recommended in the USLE (Wischmeier and Smith, 1978) or e.g. by Verstraeten et al. (2006), which might lead to estimates with large confidence intervals.

The uncertainty can be, however, reduced by combination of data from different sites (Catari et al., 2011) or by considering covariates that are better sampled or/and their variation over space and time is smaller (Goovaerts, 1999). For the spatial

interpolation of the R-factor, often variables like longitude, latitude and elevation (Goovaerts, 1999; Angulo-Martínez et al., 2009) or long-term precipitation (Lee and Lin, 2014) are considered.

In addition to spatial and temporal variability, the expression for the rainfall kinetic energy (needed for estimation of the erosivity index) and spatial interpolation are also relevant sources of uncertainty for the development of an R-factor map. The rainfall kinetic energy can be estimated by a number of expressions (see e.g. van Dijk et al., 2002). Therefore, several

authors assessed the effect on the estimated R-factor. For instance, Catari et al. (2011) mentioned variation due to kinetic energy calculation of about 10%. Similarly, many spatial models can be used to predict the R-factor values over the area. The differences between several spatial interpolation methods have been reported e.g. by Angulo-Martínez et al. (2009). Catari et al. (2011) compared the contribution of different sources of variability to the overall uncertainty in the estimated basin average R-factor in NE Spain, concluding that while the uncertainty in annual erosivity index is dominated by the temporal variability

(explaining more than 40% of variation) for long-term R-factor the kinetic energy calculation becomes more important.

Several maps of the R-factor for Europe have been released in recent decades. For instance, Van der Knijff et al. (2000) applied simple relationships between seasonal and annual rainfall and R-factor and Panagos et al. (2015) used many high-resolution station datastes (with various record length and observation periods) to derive an R-factor map based on a Gaussian process model. The values derived for the Czech Republic from these maps range between 35 and 70 MJ ha$^{-1}$ cm h$^{-1}$.

A number of estimates of the R-factor have been also published specifically for the Czech Republic. See e.g. the overview by Krása et al. (2014), who mention also results from a number of national projects. Official guidelines for soil erosion risk assessment recommend the value 20 MJ ha$^{-1}$ cm h$^{-1}$ to be used for all agricultural land in the Czech Republic. Only recently this value was increased to 40 MJ ha$^{-1}$ cm h$^{-1}$ (Janeček et al., 2007, 2012a). These values are relatively small with respect to the neighboring countries and some of the research published for the area. For instance, Janeček et al. (2006) published values

of R-factor in the range of 43–106 MJ ha$^{-1}$ cm h$^{-1}$ and concluded that considerably larger values of R-factor (45–60 MJ ha$^{-1}$





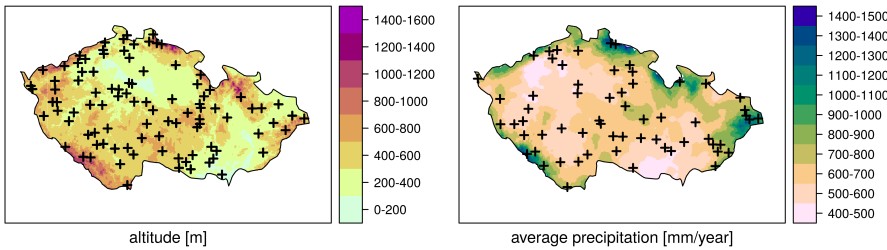

**Figure 1.** The map of the study area with (left) altitude and location of 96 stations used for spatial interpolation and (right) mean annual precipitation and location of additional stations used for assessment of uncertainty.

cm h$^{-1}$) should be used for practical application of USLE in the Czech Republic (instead of 20 MJ ha$^{-1}$ cm h$^{-1}$). Even larger values of R-factor are reported by Krása et al. (2015). On the other hand, Janeček et al. (2012b), using a regression between daily erosion index ($EI30$) and daily precipitation in order to predict annual $EI30$, report values of R-factor between 15 and 120 MJ ha$^{-1}$ cm h$^{-1}$, with an average for arable land of 30–40 MJ ha$^{-1}$ cm h$^{-1}$. Similarly, Janeček et al. (2013) derived an

R-factor for the Czech Republic from daily data considering the fraction of erosive events in each year for each station and the areal-average annual sum of the erosivity index. In addition, they excluded years with the largest and years with the smallest erosivity index from the analysis. This resulted in a recommendation to use 40 MJ ha$^{-1}$ cm h$^{-1}$ for all agricultural land in the Czech Republic. The trends in rainfall erosivity were studied by Hanel et al. (2015), who found significant a positive trend ($\approx 4\%$ per decade) in 51-year records for 11 stations (more than a half of the considered stations).

The present paper compares several methods of spatial interpolation of the R-factor over the Czech Republic and evaluates the bias and uncertainty due to expression for the kinetic energy, spatial model, record length and spatial coverage. The paper is structured as follows. The study area and data considered for calculation and spatial interpolation of the R-factor are given in Section 2. Section 3 describes the methods used for estimation of the R-factor, spatial interpolation and uncertainty assessment. The results are presented and discussed in Section 4. The paper is closed with concluding remarks (Section 5).

## 2   Study area and data

### 2.1   Study area

Because of relatively complex orography and combination of Atlantic, Mediterranean and continental effects, the precipitation patterns over the area of the Czech Republic are rather variable (see Fig. 1). The precipitation is mostly due to enhanced westerly flows in winter (Atlantic influence), except for the eastern part of the Czech Republic, which is typically influenced

by the Mediterranean Sea. The Mediterranean influences are also dominant over the whole area in summer (e.g. Brádka, 1972; Brázdil, 1980).



The mean annual total precipitation varies from about 400 mm in the western part of the Czech Republic up to more than 1400 mm in the mountains to the north (Tolasz et al., 2007). Almost two-thirds of the annual total falls in the warm half of the year (April–September). The maximum precipitation amounts can be quite large at short timescales, thus contributing considerably to the annual total precipitation. Štekl et al. (2001) point to historical records of 237 mm in 1 hour (25 May 1872),

345 mm in 1 day (29 July 1897), 537 mm in 3 days (6–8 July 1997) and 617 mm in 5 days (4–8 July 1997).

## 2.2 Pluviograph records

A database of 10-min digitized pluviograph records was used in our study. Since a considerable amount of precipitation falls as snow during winter, the data are in general available only for mid-May to mid-September (further referred to as year). This is, however, the period when heavy precipitation usually occurs. The data were digitized by the Czech Hydrometeorological

Institute (CHMI). Detailed identification and reconstruction of unreadable, damaged or missing records comprised part of the digitization process (see Květoň et al., 2004, for details). Consistency of the aggregated 10-min data with daily records from control ombrometers was further checked by Hanel and Máca (2014). Identified inconsistent days were marked unreliable and the years with more than 10% of unreliable data were excluded from the analysis. A set of 96 stations covering the study area for the period 1989–2003 was selected (based mainly on data availability) in order to provide reasonable spatial coverage and

record length for the spatial interpolation of rainfall erosivity. The distance between neighbouring stations is 5–42 km (17 km on average) and the stations are located at altitudes from 150 to 1118 m (450 m on average).

To analyze the temporal variability and the effect of the number of stations used for the spatial interpolation, also the longest available record (C2TREB01 - Třeboň), with 80 years of data and an additional 24 stations (usually with slightly different time period than 1989–2003), was considered (see Sect. 3.3). The location of the stations is indicated in Fig. 1.

The database of 10-min digitized pluviograph records was also used for the estimation of average at-site rainfall event characteristics (event depth, event duration, maximum 10-min intensity, mean event rate, time to peak, depth to peak and number of events), which were later linked with selected spatial interpolation models (see Hanel and Máca, 2014, for details on the considered rainfall event characteristics).

## 2.3 Spatial data

Gridded (1 km resolution) statistics of monthly precipitation provided by the CHMI were considered as covariates for the spatial interpolation models. Specifically, we used the average (May–September) precipitation ($r_{\mathrm{mea}}$), coefficient of variation of monthly (May–September) precipitation ($r_{\mathrm{cv}}$) and the mean excess above the 95% quantile of monthly (May–September) precipitation ($r_{\mathrm{p95}}$). Please note that these gridded statistics are based on much larger numbers of stations (usually more than 750 stations were available for each month and year) than were used in our study for spatial interpolation of the R-factor.



## 3 Methods

### 3.1 Rainfall erosivity factor

A standard methodology for the assessment of at-site rainfall erosivity factor (Wischmeier and Smith, 1978) was applied. A continuous 6-hour interval without precipitation is used to separate individual rainfall events. To be considered erosive, the cumulative rainfall of an event should be greater than 12.7 mm or the event should have at least one peak greater than 6.35 mm in 15 minutes. The latter criterion was modified to 8.5 mm in 20 minutes in order to match the temporal resolution of our dataset. Similar modification was reported by Meusburger et al. (2012). The proportion of erosive events that are included on the basis of the intensity criterion is, however, small.

The rainfall erosivity factor is defined as a long-term average annual sum of the event erosivity index ($EI30$ [MJ mm ha$^{-1}$ h$^{-1}$ yr$^{-1}$], Wischmeier and Smith, 1978),

$$EI30 = I_{30} \sum_i e_i v_i, \tag{1}$$

where

$$e_i = 28.3[1 - 0.52 \exp(-0.042 r_i)] \tag{2}$$

is the unit rainfall energy [MJ ha$^{-1}$ mm$^{-1}$] (van Dijk et al., 2002), $v_i$ and $r_i$ are the rainfall volume [mm] and intensity [mm h$^{-1}$] during a time interval $i$, respectively, and $I_{30}$ is the maximum rainfall intensity during a period of 30 min in the event [mm h$^{-1}$].

The unit rainfall energy in eq. 2 is from van Dijk et al. (2002), who assessed many expressions for its calculation. To assess the related uncertainty, the R-factor was also estimated using an additional 14 expressions for $e_i$ (see Appendix A for their definition).

Note, that the R-factor is further expressed in MJ ha$^{-1}$ cm h$^{-1}$ (equivalent to kJ mm m$^{-2}$ h$^{-1}$), which is the unit most often used in the Czech Republic.

### 3.2 Spatial interpolation models

Following the classification of spatial interpolation models for rainfall erosivity provided by Angulo-Martínez et al. (2009), selected local, global, geostatistical and mixed models were tested. For detailed discussion on spatial interpolation models of rainfall erosivity factor, see also Goovaerts (1997); Meusburger et al. (2012).

The local models, which described the spatial distribution of rainfall erosivity using local at-site information of R-factor, were represented by the model based on inverse distance weighting (IDW) (Angulo-Martínez et al., 2009 and Meusburger et al., 2012).





The IDW model predicts the R-factor $R(\boldsymbol{x_0})$ at a location $\boldsymbol{x_0}$ considering rainfall erosivity factor from $nn$ nearest stations by weighting the contribution of each station by the inverse distance $d_{0i}$ between station $i$ at location $\boldsymbol{x_i}$ to the location $\boldsymbol{x_0}$. This can be summarized by

$$R(\boldsymbol{x_0}) = \frac{\sum\limits_{i=1}^{nn} d_{0i}^{-r} R(\boldsymbol{x_i})}{\sum\limits_{j=1}^{nn} d_{0j}^{-r}}, \tag{3}$$

with exponent $r$ controlling the decay of weight with distance. Parameters $nn$ and $r$ were estimated during the model cross-validation (see sections 3.2.3 and 4).

Global models are based on regression between the R-factor and other covariates available for the whole spatial domain. These models were represented by generalized linear models (GLS). Selected geostatistical models mainly used semivariogram information about rainfall erosivity factor and other spatial covariates. This group consisted of simple kriging (SK), ordinary kriging (OK), simple cokriging (SC) and ordinary cokriging models (OC) (Goovaerts, 1997, 1999). The cokriging models SC and OC were tested using a set of seven rainfall event characteristics (see section 2.2) as a cokriging variate. The mixed models combine the global information about R-factor descriptors with variogram information and were represented by regression kriging models (RK; Hengl et al., 2004, 2007).

The global, geostatistical and mixed spatial interpolation models decompose the rainfall erosivity factor $R(\boldsymbol{x})$ at location $\boldsymbol{x}$ into two parts, using an additive model of general form

$$R(\boldsymbol{x}) = m(\boldsymbol{x}) + \boldsymbol{e}(\boldsymbol{x}), \tag{4}$$

where $m(\boldsymbol{x})$ represented the fixed or deterministic component of rainfall erosivity factor, and $\boldsymbol{e}(\boldsymbol{x})$ described the stochastic component of rainfall erosivity related to the residuals of the fixed component of the spatial interpolation model (Kitanidis, 1993; Goovaerts, 1999; Minasny and McBratney, 2007; Angulo-Martínez et al., 2009). Note, that the IDW model (eq. 3) can be interpreted as a special form of the model from eq. 4 with $\boldsymbol{e}(\boldsymbol{x})$ set to 0.

### 3.2.1 Fixed component

For the prediction of the fixed terms of the considered spatial interpolation models the linear model

$$m(\boldsymbol{x}) = \boldsymbol{f}(\boldsymbol{x})^{\top} \boldsymbol{\beta} \tag{5}$$

was used, with $\boldsymbol{f}$ the column vector of fixed term inputs at location $\boldsymbol{x}$ and $\boldsymbol{\beta}$ the column vector of parameters, which can be uniquely identified for each interpolation model (Kitanidis, 1993; Goovaerts, 1997; Minasny et al., 2011).

The fixed term consists of a single constant and $\beta$ equals to 1 for SK, OK, SC and OC models. The arithmetical mean was used as an estimator of expected value of rainfall erosivity in SK and SC models. The assumption of unknown constant mean of R-factor was applied for all OK and OC models (Goovaerts, 1997).



The fixed term for the rest of the models (GLS and RK) was estimated using three types of inputs:

1. location information represented by longitude ($x$), latitude ($y$) and altitude $z$,

2. spatial rainfall information expressed by the combinations of $r_{\mathrm{mea}}$, $r_{\mathrm{cv}}$ and $r_{\mathrm{p95}}$ and

3. information consisting of combinations of both previous types of spatial covariates.

All GLS and RK models take into account the covariance structure of residuals for the estimation of parameters $\boldsymbol{\beta}$. The applied estimator of $\boldsymbol{\beta}$ vector was defined as

$$\hat{\boldsymbol{\beta}} = \left(\mathbf{F}^{\top}\mathbf{K}^{-1}\mathbf{F}\right)^{-1}\mathbf{F}\mathbf{K}^{-1}\mathbf{R}, \tag{6}$$

with $\mathbf{F} = [\boldsymbol{f}(\boldsymbol{x_1}),\ldots,\boldsymbol{f}(\boldsymbol{x_n})]^{\top}$ the design matrix of fixed terms inputs, $\mathbf{R} = [R(\boldsymbol{x_1}),\ldots,R(\boldsymbol{x_n})]$ the rainfall erosivity calculated at $n$ locations, $\mathbf{K} = cov(\mathbf{R},\mathbf{R}^{\top})$ its covariance matrix and $\hat{\boldsymbol{\beta}}$ the estimate of $\boldsymbol{\beta}$ (Kitanidis, 1993; Goovaerts, 1999; Minasny and McBratney, 2007; Angulo-Martínez et al., 2009).

The inverse of covariance matrix $\mathbf{K}^{-1}$ was calculated using the Cholesky factorization. For the computationally unstable cases, the constant diagonal terms were added to $\mathbf{K}$ for improving its positive-definiteness (Minasny and McBratney, 2007).

### 3.2.2 Stochastic component

The random residual component $e(\boldsymbol{x})$ was expressed as a zero-mean stochastic process with the underlying isotropic spatial covariance structure $\mathbf{K} = cov(\mathbf{R},\mathbf{R}^{\top})$ for SK, OK, SC, OC and all RK models (Kitanidis, 1993; Goovaerts, 1999; Minasny and McBratney, 2007; Angulo-Martínez et al., 2009).

The rainfall erosivity $R(\boldsymbol{x_0})$ at unsampled location $\boldsymbol{x_0}$ was calculated as

$$R(\boldsymbol{x_0}) = \boldsymbol{m}(\boldsymbol{x_0}) + \boldsymbol{k}\mathbf{K}^{-1}\left(\boldsymbol{R} - \mathbf{F}\hat{\boldsymbol{\beta}}\right), \tag{7}$$

with $\boldsymbol{k} = cov(\mathbf{R}, R(\boldsymbol{x_0}))$. The second term of eq. 7 expresses the empirical best linear unbiased prediction of the stochastic component of linear mixed models (Stein, 1999; Hengl et al., 2004; Minasny and McBratney, 2007).

The covariance structure of random component was modeled using the Matérn and exponential models (Minasny and McBratney, 2005; Haskard, 2007; Pardo-Iguzquiza and Chica-Olmo, 2008; Pinheiro and Bates, 2000).

The spatial covariance was found using two approaches: 1) the classical semivariogram estimator based on the method of moments (further denoted MofM, see Goovaerts, 1997; Angulo-Martínez et al., 2009) and 2) the REML approach (Kitanidis, 1983; Pinheiro and Bates, 2000; Todini, 2001; Minasny and McBratney, 2007).

The MofM was based on calculating the semivariogram cloud of residuals and fitting of the Matérn/exponential semivariogram to the pairs of averaged semivariances calculated within each pair of two successive separation distances (Goovaerts, 1997). The semivariogram fitting often requires expert knowledge (Angulo-Martínez et al., 2009), and therefore different settings of separation distances for averaging the semivariances and reducing the effects of outliers were tried. This approach to estimating of the rainfall erosivity covariances was applied for the SK, OK, SC, OC and all regression kriging methods.





The REML approach was applied in parameter estimation for all tested GLS models. The GLS models with the exponential and Matérn (further distinguished as $GLS_M$) form of spatial covariance structure were considered. We also studied heteroscedasticity of the GLS model residuals using approach based on exponential variance function (see. 206 in Pinheiro and Bates, 2000).

### 5   3.2.3   Model selection

Standard leave-one-out procedure (Minasny and McBratney, 2007; Angulo-Martínez et al., 2009) was used for the validation of spatial interpolation models. The following indices were considered:

**Willmott's agreement index (*WI*)**

$$WI = 1 - \frac{\sum\limits_{i=1}^{n}[R_S(\boldsymbol{x_i}) - R_I(\boldsymbol{x_i})]^2}{\sum\limits_{i=1}^{n}(|R_S(\boldsymbol{x_i}) - \overline{R}_S(\boldsymbol{x_i})| + |R_I(\boldsymbol{x_i}) - \overline{R}_I(\boldsymbol{x_i})|)^2}, \tag{8}$$

**Mean absolute error (*MAE*)**

$$MAE = \frac{1}{n}\sum_{i=1}^{n}|R_S(\boldsymbol{x_i}) - R_I(\boldsymbol{x_i})|, \tag{9}$$

**Relative mean absolute error (*rMAE*)**

$$10 \qquad rMAE = \frac{1}{n}\sum_{i=1}^{n}\frac{|R_S(\boldsymbol{x_i}) - R_I(\boldsymbol{x_i})|}{R_S(\boldsymbol{x_i})}, \tag{10}$$

**Root mean square error (*RMSE*)**

$$RMSE = \sqrt{\frac{1}{n}\sum_{i=1}^{n}[R_S(\boldsymbol{x_i}) - R_I(\boldsymbol{x_i})]^2}, \tag{11}$$

where $R_S(\boldsymbol{x_i})$ is the rainfall erosivity calculated for station $i$, $R_I(\boldsymbol{x_i})$ is the spatially interpolated value of the rainfall erosivity for the same station, and $\overline{R}_S(\boldsymbol{x_i})$ and $\overline{R}_I(\boldsymbol{x_i})$ are the mean rainfall erosivities calculated for the station data and obtained from spatial interpolation, respectively.

15   The choice of the validation criteria follows the discussion about *RMSE* and *MAE* presented by Willmott and Matsuura (2005), who recommended the use of *MAE* for the estimation of average model error over the *RMSE*, since *RMSE* can be influenced by outlying observations. The *WI* was also considered for consistency with other relevant studies (for example, see Angulo-Martínez et al., 2009).

The model selection aimed at identification of robust spatial interpolation methods from the considered groups and also
20   at identification of optimal structure and parameters of the particular models including various combinations of covariates, covariance models, number of nearest stations considered for the IDW and SK, OK and RK models, etc.



### 3.3 Uncertainty assessment

The uncertainty in the derived R-factor map originates from the formulation of the kinetic energy term, spatial interpolation model and spatial and temporal variability. While the effect of the former two can be assessed by direct comparison of results for different expressions of rainfall kinetic energy and different spatial interpolation models, the effect of spatial and temporal

variability is evaluated here by simple bootstrap resampling procedures, which are briefly summarized in the rest of this section.

#### 3.3.1 Temporal variability

Record length influences the width of the confidence interval around the R-factor estimate. Due to the temporal variability, sufficient record length is required in order to provide an estimate of the R-factor such that the long-term average R-factor (further denoted the "true R-factor") would be covered by the estimated confidence interval. Therefore, we derived the confi-

dence intervals for various record lengths together with the probability that the "'true R-factor" lies within the corresponding confidence interval (further denoted the "coverage probability") using our longest available record, i.e. the station C2TREB01 (Třeboň) with 80 years of data. This can be done using a nested bootstrap procedure in which a sample of required length (e.g. 10, 20, or 80 years) is drawn (with replacement) from the original record and further resampled. This allows for determination of the confidence interval and examination whether the "true R-factor" is included. To obtain the coverage probability, the

whole procedure has to be repeated many times. Here we evaluated the coverage probabilities for the record lengths 10–80 years. For details, see Appendix B.

#### 3.3.2 Record length and spatial coverage

Long records from a dense network of stations should be ideally available to derive an R-factor map. In reality, however, long records are often available only for a relatively small number of stations and a balance between record length and spatial

coverage has to be found. It is then not clear whether longer records or better spatial coverage should be preferred.

Specifically, we asked (a) what is the relationship of error in the estimated R-factor to the spatial and temporal coverage and how spatial and temporal coverage influences (b) the width of the confidence interval around the estimated R-factor and (c) the coverage probability (i.e. the probability that the estimated confidence interval includes the "true R-factor").

To be able to assess these questions, a simulation study was conducted. The procedure is fully described in Appendix C,

and here we provide only general overview. As a reference, a synthetic dataset of monthly (May–September) erosivity index ($EI30$) was created by permutation of 10 years of data available for 120 stations as follows: First, a 100-year-long sequence of months May–September was created and a random year (from the available period) was assigned to each month in each year. Data for each of the 120 stations were then rearranged according to this year–month sequence and the data were aggregated by years. This resulted in a dataset of 100 years for 120 stations. This procedure preserves the annual cycle of erosivity index

and its spatial variability, while it assumes independence of erosivity index between individual months. This dataset is further denoted as the "full dataset". Please note that although many different replications of this "full dataset" can be obtained, only





one replication is used in this study. We do not expect the results (presented further) to vary significantly when different replication is considered.

The R-factor in this "full dataset" was estimated and a simple GLS model of the form $R \sim NAVY + r_{\mathrm{mea}} + Y$ was fitted. This model was used to predict the R-factor for an additional 62 locations (coincident with real stations, but independent of the "full dataset"). This is further denoted the "validation dataset". The assessment of the effects of spatial and temporal coverage was based on a repetitive resampling of the subsets of the "full dataset", fitting a GLS model and predicting the R-factor for the "validation dataset" (for details, refer to Appendix C).

Within the simulation study, we evaluated the RMSE, the width of the 90% confidence interval and the coverage probability for record lengths of $5, 10, 15, 20, 30, \cdots, 100$ years for $10, 20, \cdots, 120$ stations. Since it has been frequently noted that areal averaging is in general preferred to spatial interpolation in the Czech Republic (Janeček et al., 2006, 2012b, 2013), at least for agricultural areas (in general low-altitudes), we also assessed this "areal-average model" (i.e. constant R-factor for all locations). The "areal-average model" was applied for the whole "validation dataset" and considering only the stations with altitude below 600 m.

### 3.3.3 Comparison of uncertainties

To compare the contribution of different sources of uncertainty, the R-factor was predicted for the "validation data" using:

- the best GLS model of the R-factor considering 15 different expressions for kinetic energy (see Appendix A)

- selected classes of spatial interpolation models

- GLS models based on different number of stations and years (see Sect. 3.3.2)

The coefficient of variation $CV$ for each set of the estimated R-factors was calculated to summarize the variability due to different sources, similarly as done by Catari et al. (2011) or Panagos et al. (2015). For comparison, we also evaluated $CV$ for the R-factor estimates considering various record lengths for the C2TREB01 (Třeboň) station (see Sect. 3.3.1). Finally, $CV$ for the R-factor estimate for each of the 96 stations (representing natural variability) was evaluated using a simple bootstrap resampling of the annual erosivity values.

## 4 Results and discussion

### 4.1 R-factor

The estimated R-factor (considering the kinetic energy relationship proposed by van Dijk et al., 2002, eq. 2) for the 96 stations used for spatial interpolation ranges between 32 (U1KOPI01 - Kopisty, NW Czech Republic) and 152 (O1RASK01 - Raškovice, NE Czech Republic) MJ ha$^{-1}$ cm h$^{-1}$, and averages 64 MJ ha$^{-1}$ cm h$^{-1}$. Figure 2 shows average R-factor values for subsets of stations based on maximum station elevation included in the subset. For instance, the average R-factor for the





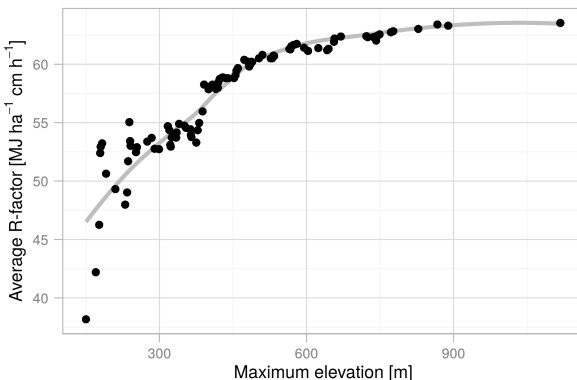

**Figure 2.** Average R-factor for subsets of stations with smaller elevation than that plotted on horizontal axis. The leftmost point corresponds to a station with the lowest altitude, the rightmost point to the overall average R-factor.

elevations up to 300 m is slightly less than 55 MJ ha$^{-1}$ cm h$^{-1}$ and for elevations up to 600 m slightly more than 60 MJ ha$^{-1}$ cm h$^{-1}$.

These values correspond well with those published by Janeček et al. (2006) and Krása et al. (2015), but are considerably larger than values recommended by the official guidelines for the Czech Republic (Janeček et al., 2007, 2012a) and those pub-

5   lished by Janeček et al. (2012b, 2013). In part, differences might be due to the modifications to the standard USLE methodology considered by Janeček et al. (2012b, 2013), e.g. calculation of the R-factor as a trimmed mean (excluding years with the two smallest and two largest annual erosivity values) of the annual erosivity index. Perhaps in small part measure, the differences can be also attributed to the different time period used for R-factor assessment (here 1989–2003, in other studies often the series from 1960 and earlier were considered).

## 10   4.2   Model selection

The estimated at-site R-factor is positively correlated with average precipitation ($r_{\mathrm{mea}}$), coefficient of variation of monthly precipitation ($r_{\mathrm{cv}}$) and the mean excess above the 95% quantile of monthly precipitation ($r_{\mathrm{p95}}$) derived from the gridded data (see Sect. 2.3), with correlation coefficients 0.75, 0.44 and 0.54, respectively. Further, positive correlation was also found with altitude (0.32) and longitude (0.47) and weak negative correlation (-0.25) with latitude. These variables have been primarily

15   used also as covariates in the relevant spatial interpolation methods.

To explore these relationships further, we analyzed a set of GLS models with various combinations of fixed component covariates and spatial stochastic covariance structure. The number of fixed term covariates ranged from one to five. Two theoretical models of spatial covariance and a heteroscedastic error model were considered (see section 3.2). The best GLS model according to the cross-validation results of *WI*, *RMSE* and *rMAE* had four fixed term covariates: $r_{\mathrm{mea}}$, $r_{\mathrm{p95}}$, altitude, and

20   longitude, exponential spatial correlation structure and exponential heteroscedastic error. The coefficient of determination for





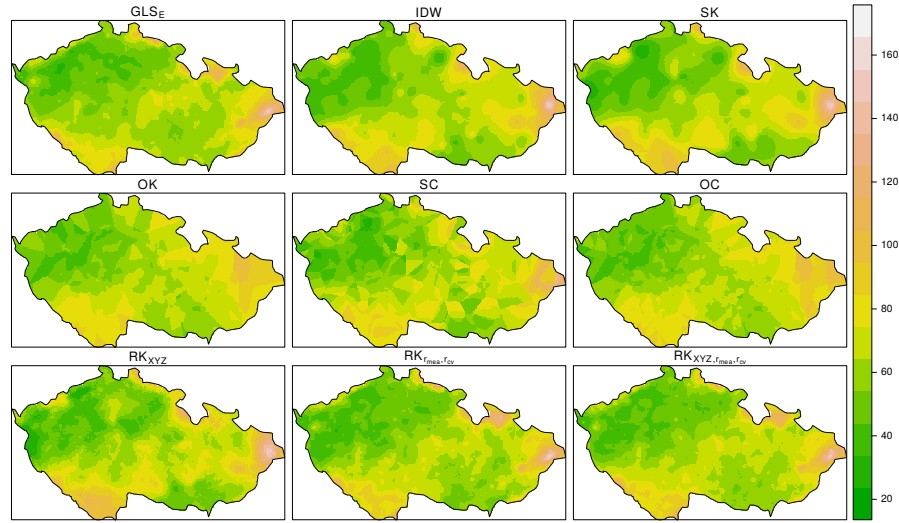

**Figure 3.** R-factor estimated by different spatial interpolation models: $\text{GLS}_\text{E}$ is represented with $R \sim r.mea + r.p95 + Y + NAVY$, where $XYZ$ stands for longitude, latitude and altitude.

regression between at-site R-factor values and R-factor values predicted by this GLS model equals to 0.7, which is comparable to results of Meusburger et al. (2012). This was followed by a GLS model with fixed term covariates longitude, latitude, altitude, $r_{mea}$ and $r_{cv}$ and Mathérn spatial covariance structure, which had the lowest value of *MAE*. These two GLS models are further referred to as $\text{GLS}_\text{E}$ and $\text{GLS}_\text{M}$, respectively.

In addition to GLS models, other spatial interpolation methods were considered. Table 1 presents cross-validation indices, spatial average and standard deviation of interpolated R-factor. Figure 3 shows the estimated R-factor for each spatial interpolation model. The estimated average R-factor ranges from 62.62 to 65.66 MJ ha$^{-1}$ cm h$^{-1}$.

Comparing the R-factor estimated by different spatial interpolation models, the largest similarities were found between the $\text{GLS}_\text{E}$, $\text{GLS}_\text{M}$ and $\text{RK}_{\text{xyz},\text{r}_{\text{mea}},\text{r}_{\text{cv}}}$. $\text{GLS}_\text{M}$ and $\text{RK}_{\text{xyz},\text{r}_{\text{mea}},\text{r}_{\text{cv}}}$ models had fixed terms inputs formed from longitude, latitude,

altitude, $r_{mea}$ and $r_{cv}$. The correlation coefficient 0.99 between estimates of $\text{GLS}_\text{E}$ and $\text{GLS}_\text{M}$ and 0.97 between the $\text{GLS}_\text{E}$ model and $\text{RK}_{\text{xyz},\text{r}_{\text{mea}},\text{r}_{\text{cv}}}$ models. These similarities were confirmed on the rasters of differences between values of the $\text{GLS}_\text{E}$ model and remaining spatial interpolation models (see Fig. 4). The median absolute difference between the $\text{GLS}_\text{E}$ and $\text{GLS}_\text{M}$ models was 2.00 MJ ha$^{-1}$ cm h$^{-1}$, with standard deviation of differences 1.92 MJ ha$^{-1}$ cm h$^{-1}$. For the differences between the $\text{GLS}_\text{E}$ and the best RK model, the median absolute difference and standard deviation was roughly double.

The largest differences were found between the $\text{GLS}_\text{E}$ model and the spatial interpolation models, which did not take into account the long-term rainfall characteristics. The largest difference of R-factor estimates (74.12 MJ ha$^{-1}$ cm h$^{-1}$) was found between the $\text{GLS}_\text{E}$ and OK models. Large values of *MAE*, *rMAE* and *RMSE* and small values of *WI* for IDW, SK, OK, SC and



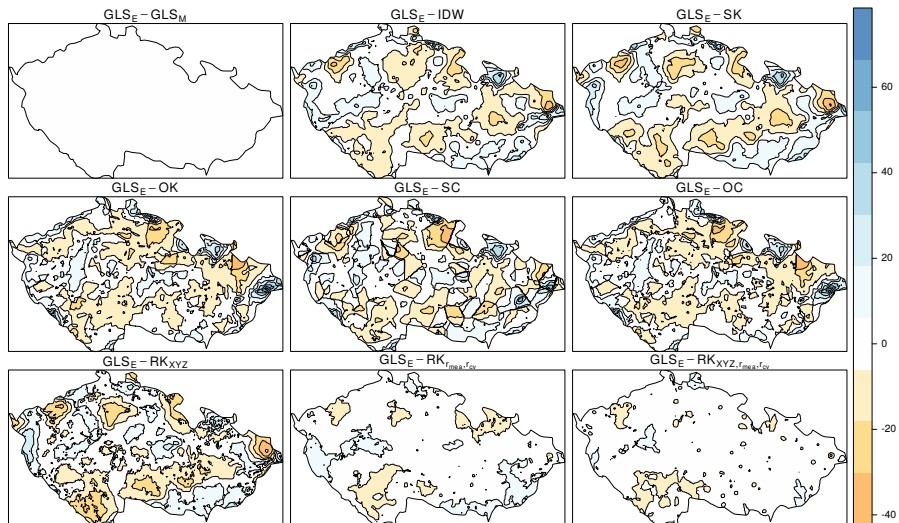

**Figure 4.** Differences [MJ ha$^{-1}$ cm h$^{-1}$] between the GLS$_E$ and other spatial interpolation models.

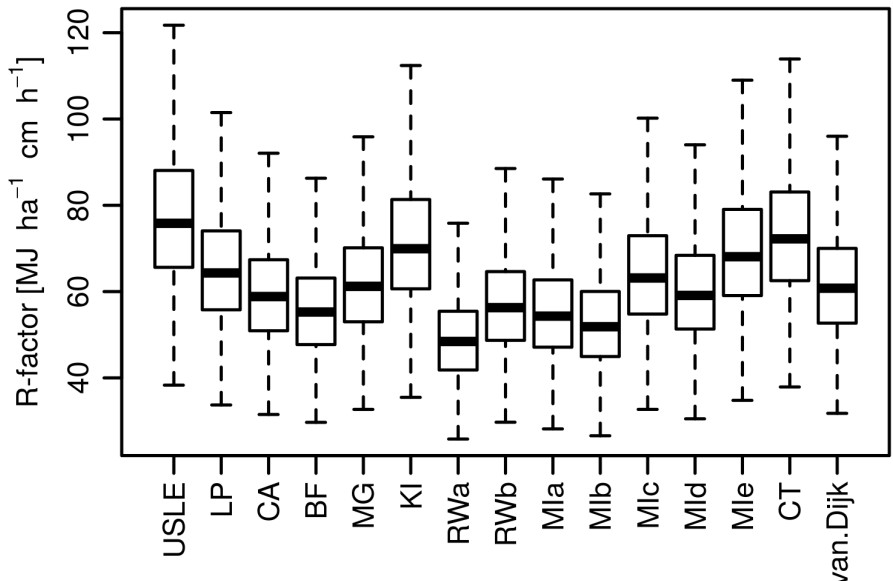

**Figure 5.** Estimated R-factor [MJ ha$^{-1}$ cm h$^{-1}$] considering different kinetic energy formulations (see Appendix A).

OC models show that spatial distribution of R-factor could not be sufficiently described by the models, which emphasized the



**Table 1.** Cross-validation indices for the best variants of each spatial interpolation model. AVst - at-site average; SDst - at-site standard deviation; AVsp - spatial average; SDsp - standard deviation of spatial R-factor. The best cross-validation indices are marked in bold font.

| | WI | RMSE | MAE | rMAE | AVst | SDst | AVsp | SDsp |
|---|---|---|---|---|---|---|---|---|
| IDW | 0.78 | 16.43 | 12.37 | 0.20 | 62.57 | 16.21 | 64.95 | 16.54 |
| SK | 0.79 | 16.30 | 12.48 | 0.21 | 62.70 | 16.37 | 64.73 | 17.45 |
| OK | 0.79 | 16.60 | 12.65 | 0.21 | 63.07 | 17.46 | 65.66 | 13.57 |
| SC | 0.73 | 16.97 | 13.12 | 0.22 | 63.13 | 13.98 | 64.45 | 15.16 |
| OC | 0.79 | 15.88 | 12.19 | 0.21 | 63.29 | 15.89 | 65.63 | 13.81 |
| $RK_{XYZ}$ | 0.79 | 17.14 | 13.13 | 0.22 | 63.69 | 19.15 | 65.37 | 18.78 |
| $RK_{r_{mea},r_{cv}}$ | 0.89 | 12.69 | 9.36 | 0.15 | 63.69 | 18.57 | 62.62 | 16.81 |
| $RK_{xyz,r_{mea},r_{cv}}$ | 0.90 | 12.19 | 9.31 | 0.15 | 63.62 | 18.37 | 63.28 | 15.88 |
| $GLS_M$ | 0.90 | 11.63 | **8.98** | **0.14** | 63.39 | 17.58 | 63.28 | 15.88 |
| $GLS_E$ | **0.91** | **11.44** | 9.00 | **0.14** | 63.56 | 17.75 | 62.78 | 15.33 |

stochastic component of R-factor, or explain R-factor using local information. The spatial interpolation models with fixed term covariates based on $r_{mea}$ or $r_{cv}$ or $r_{p95}$ were superior to the models without fixed component linked to the long-term rainfall characteristics (cf. the cross-validation results of $RK_{XYZ}$).

Including the stochastic information obtained from rainfall event characteristics in simple cokriging and ordinary cokriging

models also did not improve the spatial interpolation of R-factor. The presented SC and OC models were selected from seven different types of cokriging models. They differed according to the rainfall event characteristic, which was used for the second cokriging variate. The best SC and OC models were those, which linked their stochastic component with the maximum 10-min intensity and on site R-factor. Including these rainfall event characteristics did not, however, improve the spatial interpolation of R-factor over the spatial models based on the long-term rainfall event characteristics (seeTable 1).

The success of models including long-term precipitation characteristics might be surprising in part because daily or subdaily rainfall data are in general preferred for calculation of the R-factor over monthly or annual data (Angulo-Martínez et al., 2009). The potential of long-term rainfall characteristics for R-factor estimation is also stressed by Lee and Lin (2014), who explored relationships between rainfall and erosivity index at daily, monthly and annual time-scales and concluded that the relationship between annual erosivity index and annual rainfall is closer than that for the other time-scales.

The average R-factor varies considerably when different formulas for kinetic energy are considered (see Fig. 5). The smallest average R-factor is obtained by the RWa relationship (50 MJ ha$^{-1}$ cm h$^{-1}$) and the largest value by the standard USLE (79 MJ ha$^{-1}$ cm h$^{-1}$). The average (64 MJ ha$^{-1}$ cm h$^{-1}$) corresponds well with the value estimated using the van Dijk formula (difference is 0.6 MJ ha$^{-1}$ cm h$^{-1}$). The range between estimates for individual stations is proportional to the estimated R-factor (corresponding roughly to 40%). This also has an effect on the estimated spatial distribution of the R-factor values.





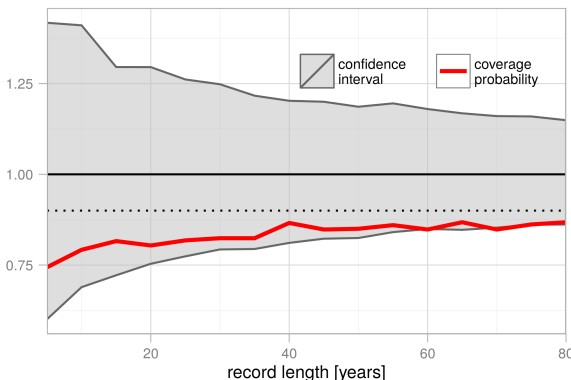

**Figure 6.** The average confidence intervals (relative to the long-term mean R-factor, 66.9 MJ ha$^{-1}$ cm h$^{-1}$; gray area) and the coverage probability (red line) for different record lengths based on the station C2TREB01 (Třeboň). The dotted line corresponds to the coverage probability of 90%.

## 4.3 Temporal variability

Wischmeier and Smith (1978) used a 22-year record to derive the rainfall erosivity factor because of "apparent cyclical patterns in rainfall data". The same is repeated by Renard et al. (1997) with a remark that longer records are advisable especially in the case that coefficient of variation of annual precipitation is large. This recommendation is often mentioned in rainfall erosivity

5   studies. However, due to data availability, shorter records are often considered at least in addition to longer records (e.g. Angulo-Martínez et al., 2009; Meusburger et al., 2012; Oliveira et al., 2013; Lee and Lin, 2014; Panagos et al., 2015). For a 105-year record from Belgium, Verstraeten et al. (2006) tested whether rainfall erosivity derived from running 10- and 22-year averages is significantly different than that form using the overall (105-year) mean. They concluded that while a 22-year period is sufficient, reliable estimates of R-factor cannot be based on 10 years of data. Apart from their study, the actual effect of the

10  sample size on the estimate of the R-factor was seldom investigated.

Fig. 6 gives the estimated confidence intervals (gray area) together with the coverage probability, i.e. the probability that the long-term mean R-factor (66.9 MJ ha$^{-1}$ cm h$^{-1}$) lies within the confidence intervals for record lengths between 10 and 80 years. The 90% confidence interval for record length 10 years ranges from 40 to 95 MJ ha$^{-1}$ cm h$^{-1}$ (i.e. $\pm$ 40%), narrows to 50–87 MJ ha$^{-1}$ cm h$^{-1}$ ($\pm$ 25–30%) for 20 years, 53–83 MJ ha$^{-1}$ cm h$^{-1}$ ($\pm$ 20–25%) and remains relatively wide, i.e. 58–77

15  MJ ha$^{-1}$ cm h$^{-1}$ ($\pm$ 15%) even for an 80-year record. The coverage probability of the 90% confidence interval is around 75% for 10-year record and from 82% for 15-year record it increases only slowly with increasing record length up to 87% for the 80-year record.





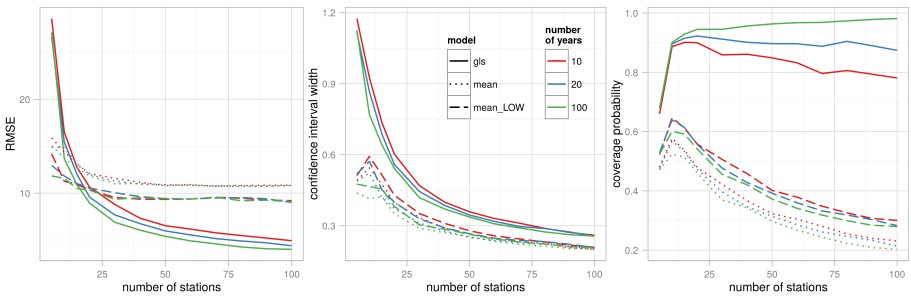

**Figure 7.** The RMSE, width of the confidence interval and the coverage probability for a GLS model (solid lines) and the "areal-average" model considering all stations (dotted lines) and only those below 600 m (dotted lines) based on simulated data with various record lengths and number of stations.

## 4.4 Spatial and temporal coverage

Fig. 7 shows the average RMSE, coverage probability and relative width of the 90% confidence interval for the validation data in cases of the GLS model (solid line), "areal-average" model (dotted line) and "areal-average" model considering only stations with altitude less than 600 m (dashed line). In the case of the GLS model, the RMSE for small numbers of stations and years is relatively large ($\approx 30$ MJ ha$^{-1}$ cm h$^{-1}$). However, it quickly drops to $\approx 10$ MJ ha$^{-1}$ cm h$^{-1}$ for 25 stations and then decreases almost linearly to $\approx 5$ MJ ha$^{-1}$ cm h$^{-1}$ for 100 stations. The RMSE for both "areal-average" models depends on the number of stations only up to 25 stations and remain almost constant for larger number of stations ($\approx 11$ MJ ha$^{-1}$ cm h$^{-1}$ when validated against all stations and $\approx 10$ MJ ha$^{-1}$ cm h$^{-1}$ when only stations below 600 m are used). The RMSE depends only slightly on the number of years used for estimation of the R-factor in the case of all models (GLS and both "areal-average" models). With respect to RMSE, the "areal-average" model is beneficial only when less than 20 stations are available. The RMSE for individual stations might be considerably larger. The 90% quantile RMSE (not shown) is around 60 MJ ha$^{-1}$ cm h$^{-1}$ for both "areal-average" models (mostly independently of the number of stations) and from 200 MJ ha$^{-1}$ cm h$^{-1}$ (10 stations) to 45 MJ ha$^{-1}$ cm h$^{-1}$ (30 stations) and 25 MJ ha$^{-1}$ cm h$^{-1}$ (100 stations) for the GLS model with 15 years of data.

The width of the 90% confidence interval (Fig. 7, middle panel) for the GLS model drops from 120% (5 stations) to 50% (25 stations) and 30% (100 stations). This value corresponds well with the width of the confidence interval for the full record of the C2TREB01 (Třeboň) station (see Fig. 6). The confidence interval for both "areal-average" models is approximately half that for the GLS model. The impact of the record length is small.

The coverage probability (Fig. 7, right panel) for the GLS model increases from 70 to 90% for 5 and 10 stations, respectively. The coverage probability further increases only when more than 20 years of data are used. For shorter records, the coverage probability decreases to $\approx 80\%$ for 100 stations. This is a consequence of faster reduction of the confidence interval width when compared to the decrease of the RMSE. Similarly, the coverage probability decreases for both "areal-average" models, since the RMSE is almost constant for more than 25 stations while the width of the confidence interval decreases. As with the RMSE, the coverage probability might be considerably lower for individual stations, especially in the case of both "areal-





**Table 2.** The coefficient of variation ($CV$) for the R-factor estimated considering different formulas for kinetic energy, spatial interpolation methods, record length and spatial coverage. In the second column, the average $CV$ for all considered stations is given, the last two columns indicate the range of the $CV$ from all stations in the "validation dataset".

| Source | Coefficient of variation [%] | | |
|---|---|---|---|
| Kinetic energy | 12.7 | (11.8 | - 14.4) |
| Spatial interpolation | 9.4 | (1.5 | - 29.7) |
| Number of stations (GLS model, *15 years*) | | | |
| *10 stations* | 31.0 | (19.6 | - 69.1) |
| *30 stations* | 10.3 | (7.6 | - 21.4) |
| *50 stations* | 7.6 | (6.0 | - 13.9) |
| Number of years (GLS model, *100 stations*) | | | |
| *5 years* | 13.0 | (10.9 | - 22.3) |
| *15 years* | 6.4 | (4.9 | - 11.8) |
| *50 years* | 4.9 | (3.4 | - 9.5) |
| Number of years (C2TREB01 - Třeboň) | | | |
| *5 years* | 37.6 | - | |
| *15 years* | 21.6 | - | |
| *50 years* | 16.1 | - | |
| *80 years* | 11.8 | - | |
| Natural variability (*96 stations*, *15 years*) | | | |
| | 23.3 | (9.1 | - 43.0) |

average" models (for a number of stations close to 0), while the coverage probability is larger than 70% for most of the stations in the case of the GLS model.

The results for all evaluated characteristics indicate that appropriate spatial coverage is more important than the length of the record, at least in situations when other relevant information to build a spatial model is available. However, at least 15–20
5  years of data should be considered (if possible) to provide reasonable coverage probabilities.

### 4.5   Comparison of different sources of uncertainty

Using the "validation data" and the GLS model, we calculated the coefficient of variation ($CV$) associated with formulation of the kinetic energy, spatial interpolation and spatial and temporal coverage (Table 2). In addition, $CV$ was also calculated for estimates of the R-factor based on different record lengths for the C2TREB01 (Třeboň) station and for the set of 96 stations
10  considered for spatial interpolation. For the latter, the estimated $CV$ was 23% on average (9–43% for all stations). This value can be interpreted as an indicator of natural variability of the R-factor based on a 15 year record. Almost the same value (21%)




is estimated for the C2TREB01 (Třeboň) station and 15-years of data. The contribution of the kinetic energy formulation ($\approx$13%) and spatial interpolation ($\approx$ 9%) is about half of this value. As expected, $CV$ of the estimates for the C2TREB01 (Třeboň) station decreases with increasing record length (38, 16 and 12% for 5, 50 and 80 years, respectively).

The same applies for the GLS model, for which, in addition, $CV$ also decreases with increasing number of stations. When comparing corresponding record lengths, the $CV$ is considerably smaller for the GLS model than for individual stations, providing a sufficient number of stations is considered in the model. For instance, for the R-factor based on 15 years of data, the average $CV$ is 31, 10, 7 and 6.4% for 10, 30, 50 and 100 stations, respectively, while for the station data and same record length the average $CV$ was 23%. In addition, considering 100 stations $CV$ is only 13(5)% for 5(50) years. From Table 2 it is evident that not only the average $CV$ decreases but the same holds also for the stations with maximum variation. For instance, using 100 stations and 15 years, the maximum $CV$ in the "validation set" is 12% for a GLS model and 43% for the station data. As the $CV$ decreases with more stations and longer records, the relative importance of the expression for the kinetic energy and spatial interpolation increases. The assessment of the sources of uncertainty could be done more formally using an analysis of variance (ANOVA) model (e.g. Yip et al., 2011) or slightly more flexible linear mixed-effects model (see e.g. Hanel and Buishand, 2015).

# 5   Summary and conclusions

In the present paper we estimated the rainfall erosivity factor (R-factor) for the area of the Czech Republic. The at-site values of the R-factor based on a 15-year record for 96 stations were considered in several spatial models in order to provide estimates of the R-factor for the whole area of the Czech Republic.

The spatial interpolation models included inverse distance weighting, simple and ordinary kriging, simple and ordinary cokriging, regression kriging with parameters estimated by the method of moments and the GLS models estimated using the REML. Several covariates have been considered to explain the spatial variation of the R-factor over the area.

In addition, uncertainty due to kinetic energy formulation, spatial model and spatial and temporal coverage was assessed by direct comparison of different methods and simulation studies.

The most important findings can be summarized as follows:

- The average R-factor in the period 1989–2003 for the considered stations is 64 MJ ha$^{-1}$ cm h$^{-1}$, with values for the individual stations between 32 and 152 MJ ha$^{-1}$ cm h$^{-1}$.

- The at-site R-factor is considerably correlated with average precipitation, coefficient of variation of monthly precipitation, the mean excess above the 95% quantile of monthly precipitation and longitude, while the correlation with altitude and latitude is weak.

- From the considered spatial models, a GLS model with altitude, latitude, mean precipitation and the mean excess above the 95% quantile of monthly precipitation provided the best performance according to three of four cross-validation indices.



- With respect to the cross-validation statistics, the spatial interpolation models that included long-term rainfall characteristics performed considerably better than those based on local interpolation and/or geographical information only.

- When the number of stations and years available for interpolation is small, the relative contribution of the uncertainty due to kinetic energy estimate and spatial interpolation method is small compared to that due to the choice of the stations and time period.

- Although the RMSE and confidence interval width decrease and coverage probability in general increases with record length and number of stations, reasonable estimates of R-factor may be obtained from relatively short records (e.g. 15–20 years) providing there is good spatial coverage.

- The spatial model should be in general preferred over the "areal-average", except for situations when only very short records for a small number of stations are available, or situations when appropriate covariates cannot be used.

**Appendix A: Considered formulations of rainfall kinetic energy**

For the application of the universal soil loss equation, Wischmeier and Smith (1978) derived a logarithmic relationship between rainfall intensity and kinetic energy of the form (converted to metric units):

$$e_i = 210 + 89 \log(r_i/10), \tag{A1}$$

where (as in eq. 2), $r_i$ is the rainfall intensity [mm h$^{-1}$] during time interval $i$. Note that slightly different coefficients are provided by Renard et al. (1997). The logarithmic relationship implies that there is no upper limit to kinetic energy, whereas research has suggested that a maximum value does exist (see van Dijk et al., 2002, for references). Therefore, Wischmeier and Smith (1978) considered constant rainfall kinetic energy for intensities greater than 76 mm h$^{-1}$. Other authors used a relationship of the form

$$e_i = e_{\max}[1 - a \exp(-b r_i)], \tag{A2}$$

where $e_{\max}$ denotes the maximum kinetic energy contents and $a$ and $b$ are empirical constants. Many different combinations of the parameters $e_{\max}$, $a$ and $b$ have been published. Van Dijk et al. (2002) therefore proposed a relationship given in eq. 2 as one providing estimates that are close to the average of many formulas for calculation of $e_u$. In the present paper, in addition to formulas given in eq. 2 and eq. A1, which are further referred to as "van Dijk" and "USLE", respectively, we calculated the $e_i$ considering a set of coefficients for eq. A2 given in table 3.


**Table 3.** Coefficients for calculation of rainfall kinetic energy in eq. A2. The acronyms used throughout the paper are given in the first column.

|  | $e_{\max}$ | $a$ | $b$ |  |
| --- | --- | --- | --- | --- |
| LP | 28.9 | 0.54 | 0.059 | Laws and Parsons (1943) |
| CA | 28.0 | 0.76 | 0.090 | Carter et al. (1974) |
| BF | 29.0 | 0.72 | 0.050 | Brown and Foster (1987) |
| MG | 29.0 | 0.72 | 0.082 | McGregor et al. (1995) |
| KI | 29.3 | 0.28 | 0.018 | Kinnell (1981) |
| RWa | 26.4 | 0.67 | 0.035 | Rosewell (1986) |
| RWb | 28.1 | 0.60 | 0.040 | Rosewell (1986) |
| MIa | 24.6 | 0.46 | 0.037 | McIsaac (1990) |
| MIb | 29.2 | 0.51 | 0.011 | McIsaac (1990) |
| MIc | 28.8 | 0.45 | 0.033 | McIsaac (1990) |
| MId | 25.1 | 0.40 | 0.045 | McIsaac (1990) |
| MIe | 26.8 | 0.29 | 0.049 | McIsaac (1990) |
| CT | 35.9 | 0.56 | 0.034 | Coutinho and Tomás (1995) |

## Appendix B: A resampling scheme for the assessment of temporal variability

Here we describe a nested bootstrap procedure in which samples of required length $l$ (e.g. 10, 20, 80 years) are repeatedly drawn from the original (observed) record and resampled to obtain the R-factor estimate with corresponding confidence interval. Finally, the probability that the "true R-factor" (here the estimate based on the full record) lies within the confidence interval is estimated.

The resampling is performed in the following steps:

1. choose the number of bootstrap samples for derivation of the confidence intervals ($n_{CI}$) and the number of bootstrap samples for the assessment of the coverage probability ($n_{CP}$); in our study we set $n_{CI} = n_{CP} = 500$

2. draw a sample of length $l$ with replacement from the original series of annual erosivity and denote this sample $s$

3. draw a sample of length $l$ with replacement from $s$ and use it to calculate the average erosivity (i.e. the R-factor)

4. repeat the previous step $n_{CI}$ times

5. calculate the 90% confidence interval from the $n_{CI}$ estimates of the R-factor from step 3 and check whether this interval includes the true R-factor

6. repeat steps 2–5 $n_{CP}$ times




7. calculate the coverage probability associated with the record length $l$ as the proportion of cases when the confidence interval from step 5 included the true R-factor

8. repeat the whole process for different record lengths $l$

Please note that the described procedure provides $n_{CP}$ estimates of the confidence intervals for specific $l$ and only their average is presented in the paper. Setting $n_{CP} = 1$ might be sufficient in the situation that only the confidence intervals would be of interest.

**Appendix C: A resampling scheme for the assessment of spatial and temporal coverage**

The following scheme describes a nested bootstrap procedure for assessment of the RMSE, coverage probability and the width of the 90% confidence interval for a GLS model considering different length of the precipitation data and different number of stations. The assessment is based on resampling of a synthetic dataset of 100 years for 120 stations (denoted "full dataset") and validated against the independent "validation dataset" (see Sect. 3.3.2). The procedure is summarized as follows:

1. draw a sample of $nyr$ years for $nsta$ stations from the "full dataset" and calculate the R-factor for each station

2. fit a GLS model of the form $R \sim NAVY + r.mea + Y$ using the sample from the previous step

3. simulate data for the given $nsta$ stations from the fitted model (see e.g. Pinheiro and Bates, 2000)

4. refit the model and use this refitted model to predict the R-factor for the "validation dataset"

5. repeat the previous two steps 500 times

6. calculate the 90% confidence interval around the estimated R-factor and the RMSE for each station of the "validation dataset" from the 500 samples obtained in steps 2–5

7. repeat the previous steps (1–6) 500 times

8. repeat the whole procedure for different $nyr$ and $nsta$

The RMSE, confidence interval and coverage probability for the "areal-average" model used for comparison was derived by replacing the estimates from the refitted model in step 4 by the areal-average of the simulated data from step 3.

*Acknowledgements.* The research has been conducted within the framework of the project "Erosion runoff - increased risk of the residents and the water quality exposure in the context of the expected climate change" (VG20122015092) sponsored by the Ministry of the Interior of the Czech Republic. Data have been kindly provided by the Czech Hydrometeorological Institute. All calculations and plotting were done in R.



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
