# Peer review of "Rainfall erosivity factor in the Czech Republic and its uncertainty"

_Hydrology and Earth System Sciences, 2016_

## Referee Comment (RC1) · Anonymous Referee #1 · 17 May 2016

Rainfall Erosivity in Czech Republic.

This is a very interesting paper as it discuss the development of rainfall erosivity dataset based on high temporal resolution data at a national level. Moreover, the manuscript has presented the main sources of uncertainty in R-factor maps. There are some issues to be addressed and reviewed before the manu8script gets published.

Find below the main points to be corrected: 1. Measurement Unit: R-factor is further expressed in MJ ha$-1$ cm h$-1$ (equivalent to kJ mm m$-2$ h$-1$), which is the unit most often used in the Czech Republic. However, here you address your result to the International public and you should adapt it to the most used measurement unit (please replace cm with mm and multiply by 10).

2. Comparison with other datasets: You estimated the mean R-factor in Czech Republic around 640 MJ mm ha-1 h-1 yr-1 using 10-minutes data. The Mean R-factor in Czech Republic at 30 minutes is 524 MJ mm ha-1 h-1 yr-1 according to the rainfall erosivity map of Europe. If you take into account the calibration factor used by Panagos et al (2016) for transferring data between 10-minutes and 30-minutes is 0.8205 then both datasets (Czech republic, European) have the same Mean (640 * 0.8205 = 525). Taking into account that for the European application another dataset has been used, both results are similar. Congratulations for the results.

3. Length of time-series and short period of stations (Paragraph P2L3-13): You need to restructure this paragraph somehow. Panagos et al (2015) have used the best available high temporal resolution data at European scale and according to Table 1 in their publications the mean length period is more than 17.5 years (half of the countries had records more than 20 years). Only countries with low erosivity (Finland, Estonia, Latvia, Romania) had records covering short periods. Angulo-Martinez (2009) estimated the R-factor in Ebro based on 10 years data. So, you cannot compare the data availability in Europe (or in one country or at regional level) with the 1 station made available by Verstraeten(2006). Moreover, your estimates are based on 14 years data.

4. Whole section 3.2: You need to restructure this section by giving a short description of each model used (IDW, GLS, OK, RK. Etc) and avoid the whole part on Fixed and stochastic component. Those 2 parts seems like "technical note" from the geostatistical books. References will be enough while you have simply to describe the models used in your case. 5. Figures should be self-explained. So In Figure 3, the reader cannot understand what 'GLS" , SK, etc is. . ..Please put the explanations in the caption.

Some additional adjustments requested: P1, L17: Soil erosion by water is a widespread problem throughout Europe (Van der Knijff et al., 2000). Citation to the paper of 2000. There is a more recent and accurate development of soil erosion map in Europe (2015) and you should update this citation with the new one.

L2P28: Replace datastes with datasets.

In the introduction, I would also expect one sentence (plus necessary citations) regarding other recently developed rainfall erosivity datasets at national scale. There are new R-factor datasets for Italy, Greece, Brazil, Chile, China, Australia, New Zealand etc.

P3 L8-9: "The trends in rainfall erosivity were studied by Hanel et al. (2015), who found significant a positive trend (= 4% per decade) in 51-year records for 11 stations (more than a half of the considered stations).". Trend on what? And compared to what? Increase or decrease?

Please replace sub-title: 2.3 Spatial Data" with "Precipitation maps" or "Precipitation Spatial datasets". Spatial data can be everyting. . ...

P7L24: REML stands for??? Please explain

Figure 1: please delete the word "The map" and rephrase the caption.

P15 : Last paragraph: The ranges you presented are quite wide. I doubt that with 80-year data records the range can be +-15%.

Conclusions: what is the "areal-average". Please correct it.
* * *

---

## Referee Comment (RC2) · Anonymous Referee #2 · 3 Jun 2016

In this manuscript, the authors explored the spatial distribution of the rainfall erosivity index R in Czech Republic. The work is based on data collected from 96 stations at a 10-min time interval for the period 1989–2003. Different interpolation methods have been used in order to evaluate the spatial uncertainty related to each interpolation model. The best results have been obtained by the generalized least-squares (GLS) model. Another source of uncertainty was taken into account considering different equations used to estimate the rainfall kinetic energy (required for calculation of the R-factor). Finally, the effects of record length and spatial coverage have been considered. The authors concluded that if sufficient spatial coverage and covariates are available, reasonable estimates of the R-factor can be obtained even from relatively short records (15–20 years).

The paper seems to me well structured and the statistical analyses satisfactory. I think

the paper needs exposure at an international level but some details need to be added in this version.

Specific comments In general, I agree with the authors that the rainfall erosivity index is a good indicator of areas subjected to soil erosion. However, I do not believe that using only 14-15 years of measurements it is possible to obtain a good estimate of long-term rainfall erosivity, everywhere. The period covered by the datasets is short and may (or may not) contain outliers which generally affect the mean values. Even if some authors are inclined to remove the outliers from the series (see Janecek et al., 2013), I do not think it is a correct approach because these outliers often are the major contributors (up to 80%) of the total amount of soil eroded from an area (see for example Martinez-Casasnovas et al., 2002; Fang et al., 2013). In this respect, the authors based their long-term analysis on a single dataset (C2TREB01, with 80 years available). As I understand, this station shows a value of R = 669 (MJ ha-1 mm h-1) which falls perfectly around the mean value calculated for the entire region (ca. 640). The same consideration can be extended to the CV value (CV = 21.6 for C2TREB01, considering 15 years, and CV=23.3 for the entire region – see natural variability for 96 stations in Table 2). In other words, this station is indicative of the average conditions of the region and it is not surprising that the bootstrap analysis indicated in section 3.3.1 and appendix b gives no strong differences if periods of different length are considered. It would be interesting to look at another station, with similar length, but showing a higher variability of the rainfall erosivity factor. If the authors have this information, this can be added to improve the paper. If not, please, add some comments that emphasise this uncertainty.

During the last 2-3 decades, an increase in the rainfall erosivity factor is documented in different areas of the world due to climate change (see among the others Fiener et al., 2013; Nearing et al., 2004; Porto et al., 2013; Capra et al., 2015; Zhang et al., 2005). This is documented also by the first author in a previous contribution (see Hanel et al., 2016) for some stations in Czech Republic. I suggest to show a figure with the 80

values of R (calculated for C2TREB01) vs time (years) in order to see if no increasing trend can be detected during the period 1989-2003. If there is an increasing trend in this period, it means that the 96 values of R are not stationary and this needs some more comments.

The authors said (citing also Goovaerts, 1999; Angulo-Martínez et al., 2009) that using covariates like longitude, latitude and elevation or long-term precipitation it is possible to cover the existing gaps of direct evaluation of R. I want to emphasise here that such correlations are acceptable only where the R values are obtained using indirect methods that involve, for example, rainfall values at daily or monthly scale (see Capra et al., 2015). When short time steps are considered (and R requires time intervals shorter than 30 minutes) these correlations fail (see for example Porto, 2016), unless climatic conditions are uniform over large areas. But, as the authors recognise, R values are very much affected by local conditions and this complicates things. I am sure the authors want to add some more comments here.

References

Capra A., Porto P., and La Spada C. (2015). Long-term variation of rainfall erosivity in Calabria (Southern Italy). Theor Appl Climatol DOI 10.1007/s00704-015-1697-2

Fang N-F, Shi Z-H, Yue B-J, Wang L (2013). The Characteristics of Extreme Erosion Events in a Small Mountainous Watershed. PLoS ONE 8(10): e76610. doi:10.1371/journal.pone.0076610

Fiener P, Neuhaus P, Botschek J. (2013). Long-term trends in rainfall erosivity – analysis of high resolution precipitation time series (1937–2007) fromWestern Germany. Agric. For. Meteorol. 171–172:115–123.

Martin Hanel, Alena Pavlásková and Jan Kyselý (2016). Trends in characteristics of sub-daily heavy precipitation and rainfall erosivity in the Czech Republic. Int. J. Climatol. 36: 1833–1845

J.A. MartÄśnez-Casasnovas, M.C. Ramos, M. Ribes-Dasi (2005). Soil erosion caused by extreme rainfall events: mapping and quantification in agricultural plots from very detailed digital elevation models. Geoderma 105: 125–140

Meusburger, K., Steel, A., Panagos, P., Montanarella, L., Alewell, C. (2012). Spatial and Temporal Variability of Rainfall Erosivity Factor for Switzerland. Hydrology and Earth System Sciences 16: 167–177.

Nearing, M.A., Pruski, F.F., O'Neal, M.R. (2004). Expected climate change impacts on soil erosion rates: A review. Journal of Soil and Water Conservation 59, 43-50.

Porto P. (2016). Exploring the effect of different time resolutions to calculate the rainfall erosivity factor R in Calabria, southern Italy. Hydrol. Process. 30, 1551–1562.

Porto, P., Walling, D.E., Callegari G. (2013). Using 137Cs and 210Pbex measurements to investigate the sediment budget of a small forested catchment in Southern Italy. Hydrological Processes 27(6): 795-806.

G.-H. Zhang, M. A Nearing, B.-Y. Liu (2005). Potential effects of climate change on rainfall erosivity in the Yellow River basin of China. Transactions of the ASAE, Vol. 48(2): 511-517.
* * *

---

## Author Comment (AC1) · 10 Jun 2016

We thank very much for the constructive comments which helped to improve the manuscript. The point-by-point response is included below, the modified version of the manuscript is attached.

**Find below the main points to be corrected:**

**1. Measurement Unit: R-factor is further expressed in MJ ha1 cm h1 (equivalent to kJ mm m2 h1), which is the unit most often used in the Czech Republic. However, here you address your result to the International public and you should adapt it to the most used measurement unit (please replace cm with mm and multiply by 10).**

[Figure]

We agree, that using Czech conventional unit might be confusing for international audience. The units were changed as requested in the new version of the manuscript (including main text, tables and figures).

**2. Comparison with other datasets: You estimated the mean R-factor in Czech Republic around 640 MJ mm ha-1 h-1 yr-1 using 10-minutes data. The Mean R-factor in Czech Republic at 30 minutes is 524 MJ mm ha-1 h-1 yr-1 according to the rainfall erosivity map of Europe. If you take into account the calibration factor used by Panagos et al (2016) for transferring data between 10-minutes and 30-minutes is 0.8205 then both datasets (Czech republic, European) have the same Mean (640 \* 0.8205 = 525). Taking into account that for the European application another dataset has been used, both results are similar. Congratulations for the results.**

Thank you for this point. This infromation is now included in the main text.

**3. Length of time-series and short period of stations (Paragraph P2L3-13): You need to restructure this paragraph somehow. Panagos et al (2015) have used the best available high temporal resolution data at European scale and according to Table 1 in their publications the mean length period is more than 17.5 years (half of the countries had records more than 20 years). Only countries with low erosivity (Finland, Estonia, Latvia, Romania) had records covering short periods. Angulo-Martinez (2009) estimated the R-factor in Ebro based on 10 years data. So, you cannot compare the data availability in Europe (or in one country or at regional level) with the 1 station made available by Verstraeten(2006). Moreover, your estimates are based on 14 years data.**

Our intention was not to assert that Panagos et al. (2015) or Angulo-Martínez et al. (2009) used insufficient data for R-factor assessment, rather we tried to point out, that although the best available high temporal resolution records in some regions might be considerably shorter than the length recommended by guidelines, the R-factor often

needs to be estimated at these locations using the available data. The second point was, that in the case when only short records are available, it is possible to reduce the uncertainty by regional analysis or/and by employing appropriate covariates into the analysis. To make these points clear, the paragraph has been modified.

**4. Whole section 3.2: You need to restructure this section by giving a short description of each model used (IDW, GLS, OK, RK. Etc) and avoid the whole part on Fixed and stochastic component. Those 2 parts seems like technical note from the geostatistical books. References will be enough while you have simply to describe the models used in your case. 5. Figures should be self-explained. So In Figure 3, the reader cannot understand what GLS, SK, etc is. Please put the explanations in the caption.**

We agree, that it is not necesary to include details that can be found in statistical books, therefore the whole section 3.2 was rewritten a substantially shortened.

**Some additional adjustments requested:**

**P1, L17: Soil erosion by water is a widespread problem throughout Europe (Van der Knijff et al., 2000). Citation to the paper of 2000. There is a more recent and accurate development of soil erosion map in Europe (2015) and you should update this citation with the new one.**

The sentence was updated.

**L2P28: Replace datastes with datasets.**

This was corrected.

**In the introduction, I would also expect one sentence (plus necessary citations) regarding other recently developed rainfall erosivity datasets at national scale. There are new R-factor datasets for Italy, Greece, Brazil, Chile, China, Australia, New Zealand etc.**

[Figure]

This was added.

**P3 L8-9: The trends in rainfall erosivity were studied by Hanel et al. (2015), who found significant a positive trend (= 4% per decade) in 51-year records for 11 stations (more than a half of the considered stations). Trend on what? And compared to what? Increase or decrease?**

The information was clarified.

**Please replace sub-title: 2.3 Spatial Data with Precipitation maps or Precipitation Spatial datasets. Spatial data can be everyting.**

The section was renamed to "Gridded precipitation data".

**P7L24: REML stands for??? Please explain**

The abreviation REML - restricted maximum likelihood - is explained in a new section 3.2.

**Figure 1: please delete the word The map and rephrase the caption**

The caption was modified.

**P15 : Last paragraph: The ranges you presented are quite wide. I doubt that with 80-year data records the range can be +-15%.**

The surprisingly large width of the confidence interval for the Rfactor estimate based on 80 years of data relates to large variation of annual rainfall erosivity, which is due to its non-linear relation to rain intensity and depth. Note for instance, that for the same station and 10 year record the width of the confidence interval for 10 min annual maxima and rainfall total is 70% and 40% that for rainfall erosivity, respectively. This difference only slightly decreases for longer averaging periods. Assuming that decrease in width of the confidence interval with number of years is proportional to corresponding standard deviation and assuming independence between years, the width of the confidence interval should be inversely proportional to the square root of the number of

years. For the confidence interval around the estimated Rfactor this leads to drop from $\pm$ 40% to $\pm$ 14% $(40/\sqrt{8})$, which matches our estimate well. Note, that in the case of dependence between years, the standard deviation (and thus the confidence interval) is expected to be larger.

We added a short note on this.

**Conclusions: what is the areal-average. Please correct it.**

The sentence was modified.

Please also note the supplement to this comment:
http://www.hydrol-earth-syst-sci-discuss.net/hess-2016-158/hess-2016-158-AC1-supplement.pdf

———————————————————————

[Figure]

**Supplement:**

[revised manuscript text omitted]

---

## Referee Comment (RC3) · Anonymous Referee #3 · 23 Jun 2016

In general, this is a well structured and well written manuscript; Results are not only applicable on a local scale (CZ), they will also provide valuable input to the scientific community; At present however, the manuscript lacks detail of information which needs to be added. More specific remarks:

1) I find it impossible to recalculate any of the results obtained due to the complete lack of parameter values for the different equations tested; I suggest to add a table with parameter values whenever possible; 2) Detailed information on input data is missing (station name, exact period of recording, details about covariate values....) in addition a table with information on R-factor characteristics (mean R-factor) of the stations is missing, this may already be included into the table of input information - please provide; I am aware that these details will need about two pages of the manuscript, however without this information, the manuscript lacks much of detail. 3) Please reconsider the number of digits you are using to describe results. Given the fact that you are dealing with confidence intervals in the range of $\pm$ 10 (minimum) it does not make sense to provide R-factors with 2 digits after the decimal. See for instance page 12, line 7 or Table 1. Please reconsider throughout the whole manuscript. 4) For practical purposes (a useful application of the USLE) it will be necessary to provide at least monthly R factors, because they are needed as input into the USLE management factor. I understand that it might beyond the scope of this paper, however I would strongly suggest to provide these data in the future. 5) I am missing some information about stationarity of the data used for the study. Can you provide some information here? 6) Page 2, line 29: It is interesting to note that, while the mean R-factor values of maps based on a European dataset (Panagos et al., 2015) are quite similar to those derived in this manuscript, their range is much smaller. For the extreme case of an R factor of 152 (recorded at one site in Czech Republic) this would practically increase a soil loss according to some USLE approach for >100%. 7) Page 4, line 25: Is the gridded information data set using the same time period as the station specific data set? Please provide this information.

8) Figure 3: This Figure does not provide useful information at present – either rework for a better graphical representation or skip

9) Figure 7: . . .. only those below 600 m (dashed). . . ..

---

## Author Comment (AC2) · 24 Jun 2016

We thank to pointing us towards some further explorations and discussion. The response to the specific comments is included bellow. The modified manuscipt is attached.

**Specific comments:**

**In general, I agree with the authors that the rainfall erosivity index is a good indicator of areas subjected to soil erosion. However, I do not believe that using only 14-15 years of measurements it is possible to obtain a good estimate of long-term rainfall erosivity, everywhere. The period covered by the datasets is short and may (or may not) contain outliers which generally affect the mean values. Even if some authors are inclined to remove the outliers from the series**

[Figure]

**(see Janecek et al., 2013), I do not think it is a correct approach because these outliers often are the major contributors (up to 80%) of the total amount of soil eroded from an area (see for example Martinez-Casasnovas et al., 2002; Fang et al., 2013). In this respect, the authors based their long-term analysis on a single dataset (C2TREB01, with 80 years available). As I understand, this station shows a value of R = 669 (MJ ha-1 mm h-1) which falls perfectly around the mean value calculated for the entire region (ca. 640). The same consideration can be extended to the CV value (CV = 21.6 for C2TREB01, considering 15 years, and CV=23.3 for the entire region – see natural variability for 96 stations in Table 2). In other words, this station is indicative of the average conditions of the region and it is not surprising that the bootstrap analysis indicated in section 3.3.1 and appendix b gives no strong differences if periods of different length are considered. It would be interesting to look at another station, with similar length, but showing a higher variability of the rainfall erosivity factor. If the authors have this information, this can be added to improve the paper. If not, please, add some comments that emphasise this uncertainty.**

Thank you for pointing this out. Indeed the coverage probability and the width of the confidence interval is influenced by variability of the erosivity index at a location. Since the only station with similar record length that is available to us is located close to C2TREB01 and has similar characteristics, we demonstrate this effect using simulated data. At most sites (including C2TREB01), the annual erosivity index ($EI30$) can be described by gamma distribution (assessed by the Anderson-Darling test), i.e. $EI30 \sim \Gamma(\alpha, \beta)$, with $\alpha$ and $\beta$ the shape and rate parameter, respectively. It can be shown, that in order to change the coefficient of variation by factor $k$, the parameters have to be modified as follows:

$$\alpha^* = R \frac{\beta}{k^2}, \beta^* = \frac{\beta}{k^2}$$

(with $\alpha^*$ and $\beta^*$ the modified shape and rate), provided the R-factor ($R = $ mean $EI30$)

[Figure]

remains constant. We estimated $\alpha$ and $\beta$ using the whole record from the C2TREB01 station and modified these parameters considering $k = 0.5, 1$ and $2$. The scheme from appendix B was used to assess the coverage probability and confidence intervals, except point 2, where the sample of length $l$ was generated from the modified distribution. The results are shown in the right panel of Fig. 1. It is clear, that the confidence intervals as well as the coverage probability for $k = 1$ correspond reasonably with those from the left panel of the same figure. It is also clear (and expected) that the confidence interval width increases with coefficient of variation. For instance for 15 year record and doubling of coefficient of variation it ranges from 0.5 to 1.57. For increasing record lengths the coverage probability increases and the width of the confidence interval decreases. Note that the confidence interval for erosivity index with large coefficient of variation remains large (>50%) even for 80 years of data. The coverage probability, on the other hand, decreases only slightly with CV.

In reaction to your comment we extended Fig. 6 of the manuscript and added a paragraph at the end of section 4.3.

**During the last 2-3 decades, an increase in the rainfall erosivity factor is documented in different areas of the world due to climate change (see among the others Fiener et al., 2013; Nearing et al., 2004; Porto et al., 2013; Capra et al., 2015; Zhang et al., 2005). This is documented also by the first author in a previous contribution (see Hanel et al., 2016) for some stations in Czech Republic. I suggest to show a figure with the 80 values of R (calculated for C2TREB01) vs time (years) in order to see if no increasing trend can be detected during the period 1989-2003. If there is an increasing trend in this period, it means that the 96 values of R are not stationary and this needs some more comments.**

It is true, that positive trend was detected for some stations in the Czech Republic for the period 1961-2011 in our previous study. Looking at the period 1989-2003 no clear trend is obvious (Fig. 2). We added a sentence on that in the section 2.2.

[Figure]

**The authors said (citing also Goovaerts, 1999; Angulo-Martínez et al., 2009) that using covariates like longitude, latitude and elevation or long-term precipitation it is possible to cover the existing gaps of direct evaluation of R. I want to emphasise here that such correlations are acceptable only where the R values are obtained using indirect methods that involve, for example, rainfall values at daily or monthly scale (see Capra et al., 2015). When short time steps are considered (and R requires time intervals shorter than 30 minutes) these correlations fail (see for example Porto, 2016), unless climatic conditions are uniform over large areas. But, as the authors recognise, R values are very much affected by local conditions and this complicates things. I am sure the authors want to add some more comments here.**

The correlation between the Rfactor and various covariates is discussed in the beginning of section 4.2 (i.e. correlation between Rfactor and $r_{mea}$ is 0.75, for $r_{cv}$ it is 0.44, for $r_{p95}$ 0.54, elevation 0.32, longitude 0.49 and latitude -0.25). Fig. 3 bellow show scatter plots of Rfactor and selected covariates. The relevance of these covariates for the Rfactor is also obvious from the cross-validation of different GLS models (see tab. ) bellow. Finally, a GLS model considering $r_{mea}$ can explain around 59% variability of Rfactor, $r_{p95}$ around 31% and $r_{cv}$ or longitude around 21% of variability. The elevation is, on the other hand, rather poor covariate (when not considered in combination with some characteristics of precipitation).

However, in line with your comment, we agree that the correlation between Rfactor and long term characteristics of precipitation, its variability and topographic indices, which is considerable in our dataset, might be very different in more complex regions as documented by Capra et al. (2015) or Porto (2016). A note on this was added.

**Table 1.** Cross-validation of GLS models with different combinations of covariates.

| GLS model | WI | RMSE | MAE | rMAE | AVst | SDst |
|---|---|---|---|---|---|---|
| $R \sim r_{mea}$ | 0.83 | 139.47 | 108.74 | 0.18 | 619.06 | 146.62 |
| $R \sim r_{mea} + X$ | 0.84 | 138.80 | 106.75 | 0.17 | 600.22 | 148.66 |
| $R \sim r_{mea} + r_{p95} + Y$ | 0.90 | 116.98 | 92.62 | 0.15 | 635.57 | 175.48 |
| $GLS_E$ | 0.91 | 114.37 | 89.95 | 0.14 | 635.64 | 177.47 |
| $GLS_M$ | 0.90 | 115.38 | 90.57 | 0.15 | 635.34 | 175.12 |

Watershed. PLoS ONE 8(10): e76610. doi:10.1371/journal.pone.0076610

Fiener P, Neuhaus P, Botschek J. (2013). Long-term trends in rainfall erosivity – analysis of high resolution precipitation time series (1937–2007) from Western Germany. Agric. For. Meteorol. 171–172:115–123.

Martin Hanel, Alena Pavlásková and Jan Kyselǎ¡ (2016). Trends in characteristics of sub-daily heavy precipitation and rainfall erosivity in the Czech Republic. Int. J. Climatol. 36: 1833–1845

J.A. Martǎsnez-Casasnovas, M.C. Ramos, M. Ribes-Dasi (2005). Soil erosion caused by extreme rainfall events: mapping and quantification in agricultural plots from very detailed digital elevation models. Geoderma 105: 125–140

Meusburger, K., Steel, A., Panagos, P., Montanarella, L., Alewell, C. (2012). Spatial and Temporal Variability of Rainfall Erosivity Factor for Switzerland. Hydrology and Earth System Sciences 16: 167–177.

Nearing, M.A., Pruski, F.F., O'Neal, M.R. (2004). Expected climate change impacts on soil erosion rates: A review. Journal of Soil and Water Conservation 59, 43-50.

Porto P. (2016). Exploring the effect of different time resolutions to calculate the rainfall erosivity factor R in Calabria, southern Italy. Hydrol. Process. 30, 1551–1562.

Porto, P., Walling, D.E., Callegari G. (2013). Using 137Cs and 210Pbex measurements to investigate the sediment budget of a small forested catchment in Southern Italy. Hydrological Processes 27(6): 795-806.

G.-H. Zhang, M. A Nearing, B.-Y. Liu (2005). Potential effects of climate change on rainfall erosivity in the Yellow River basin of China. Transactions of the ASAE, Vol. 48(2): 511-517.

Please also note the supplement to this comment:
http://www.hydrol-earth-syst-sci-discuss.net/hess-2016-158/hess-2016-158-AC2-supplement.pdf

—————————————————————

[Figure]

**Fig. 1.** The average confidence intervals (relative to the long-term mean R-factor, 669 MJ ha$^{-1}$ mm h$^{-1}$; gray area) and the coverage probability (thick lines) for different record lengths based on the

**Fig. 2.** Annual erosivity index (EI30) for station C2TREB01

**Fig. 3.** Scatter plots for Rfactor and selected covariates.

**Supplement:**

[revised manuscript text omitted]

---

## Author Comment (AC3) · 29 Jun 2016

We thank for practically oriented comments. The point-by-point response is included bellow. In response to comments we replaced one figure and modified the manuscript at several places. In addition, we now include supplement providing detailed information on the considered data. The modified manuscript is attached.

**More specific remarks:**

**1) I find it impossible to recalculate any of the results obtained due to the complete lack of parameter values for the different equations tested; I suggest to add a table with parameter values whenever possible**

We agree, that for practical purposes, it is useful to provide parameters allowing for

estimation of R-factor. However, in our opinion, it is sufficient to include estimated parameters for the best interpolation model (GLS_E) only, since it is in general preferred over the other models. Note also, that the parameter matrices for some other models are huge and their practical implementation would be rather difficult. The estimated parameters for the best interpolation model (GLS_E) are now included in the revised manuscript (Sect. 4.2).

**2) Detailed information on input data is missing (station name, exact period of recording, details about covariate values.. . .) in addition a table with information on R-factor characteristics (mean R-factor) of the stations is missing, this may already be included into the table of input information - please provide; I am aware that these details will need about two pages of the manuscript, however without this information, the manuscript lacks much of detail.**

We believe, that such table is too large to fit in the manuscript. However, we agree that such detailed information might be interesting/useful for some readers. Therefore we extended the manuscript with a supplement providing this information (station identificator, name, coordinates, altitude, covariate values, the at-site R factor and the number of missing/unreliable years within the considered period 1989-2003).

**3) Please reconsider the number of digits you are using to describe results. Given the fact that you are dealing with confidence intervals in the range of $\pm$ 10 (minimum) it does not make sense to provide R-factors with 2 digits after the decimal. See for instance page 12, line 7 or Table 1. Please reconsider throughout the whole manuscript.**

In response to anonymous referee #1 we already modified units in which the R-factor is presented in the manuscript. We agree that it is sufficient to provide rounded R-factor values. This was checked throughout the manuscript.

**4) For practical purposes (a useful application of the USLE) it will be necessary to provide at least monthly R factors, because they are needed as input into the**

**USLE management factor. I understand that it might beyond the scope of this paper, however I would strongly suggest to provide these data in the future.**

We understand this point, but it is indeed out of the scope of this paper to consider monthly R-factor values. Please note, that for instance it might be good to purchase the covariate values (gridded data) for individual months. To increase a potential practical impact of our study, however, we at least provide a typical seasonal distribution of the erosivity index in the modified manuscript (see Sect. 4.1).

**5) I am missing some information about stationarity of the data used for the study. Can you provide some information here?**

The erosivity index shows no clear trend in the considered period (1989-2003). This is now noted in Sect. 2.2 (also as a reaction to anonymous referee #2).

**6) Page 2, line 29: It is interesting to note that, while the mean R-factor values of maps based on a European dataset (Panagos et al., 2015) are quite similar to those derived in this manuscript, their range is much smaller. For the extreme case of an R factor of 152 (recorded at one site in Czech Republic) this would practically increase a soil loss according to some USLE approach for >100%.**

Thank you for this point. It is true, indeed, that the range of R-factor values for the Czech Republic is narrower in the map provided by Panagos (2015) - further denoted PNGS2015 - compared to our results. This is likely due to different (and smaller) number and location of stations used for derivation of the maps. The range of the R-factor values for the Czech Republic from PNGS2015 is ca <340 - 900 [MJ ha-1 mm h-1]. After correction for temporal resolution (conversion from 30 min to 10 min, also provided in PNGS2015) it becomes ca <414 - 1097 [MJ ha-1 mm h-1]. While the maximum at-site R-factor is 1520 [MJ ha-1 mm h-1] (O1RASK01), the second largest at-site R-factor only slightly exceeds 1100 [MJ ha-1 mm h-1], corresponding well with the maximum from PNGS2015.

Note that the R-factor map, when derived without the O1RASK01 station, is very similar to the map presented in the manuscript - the maximum, mean and spatial distribution of R-factor changes only very slightly, suggesting our model is rather robust. We added a note on this. In addition, in response to comment #8 we modified Fig.3, showing now also R-factor maps based on different sets of stations.

**7) Page 4, line 25: Is the gridded information data set using the same time period as the station specific data set? Please provide this information.**

Yes, the period considered for the derivation of the gridded data is the same as for the station data. It is now stated explicitly in the manuscript (see Sect. 2.3).

**8) Figure 3: This Figure does not provide useful information at present – either rework for a better graphical representation or skip**

The figure was modified - we decreased the number of panels. The four displayed maps now show the estimated R-factor according to the "best" model (GLS$_E$) fitted on full set of 96 stations. Other 3 panels demonstrate the effect of excluding stations with large R-factor values, responding to comment #6.

**9) Figure 7: . . .. only those below 600 m (dashed).**

Thank you for spotting this error. It is now corrected.

Please also note the supplement to this comment:
http://www.hydrol-earth-syst-sci-discuss.net/hess-2016-158/hess-2016-158-AC3-supplement.zip

---

## Author Response (AR2)

Dear prof. Mikos,

Thank you very much for handling our manuscript as well as for positive evaluation and suggested corrections. They are all incorporated now.

Best regards,
Martin Hanel